# Regenerative Medicine to Improve Outcomes of Ventral Hernia Reconstruction (REPAIR Study) Phase 1: Find the Best Decellularization Protocol for the Human Dermis

**DOI:** 10.3390/jpm12091411

**Published:** 2022-08-30

**Authors:** Michele Altomare, Luca Ferrario, Laura Benuzzi, Marta Cecilia Tosca, Marta Gipponi, Imam Khodri, Giovanni Sesana, Stefania Cimbanassi, Stefano Piero Bernardo Cioffi, Andrea Spota, Roberto Bini, Osvaldo Chiara, Andrea Mingoli

**Affiliations:** 1Department of Surgical Sciences, Sapienza University of Rome, Piazzale Aldo Moro 5, 00185 Rome, Italy; 2General Surgery and Trauma Team, ASST GOM Niguarda, Piazza Ospedale Maggiore 3, 20162 Milan, Italy; 3General Surgery Residency Program, State University of Milan, Via Festa del Perdono 7, 20122 Milan, Italy; 4Tissue Bank and Tissue Therapy Unit, Emergency and Acceptance Department, ASST Niguarda Hospital, Piazza Ospedale Maggiore 3, 20162 Milan, Italy; 5Department of Pathophysiology and Transplants, State University of Milan, Via Festa del Perdono 7, 20122 Milan, Italy; 6Department of Surgery “P. Valdoni”, Sapienza University of Rome, Viale del Policlinico 155, 00161 Rome, Italy

**Keywords:** wound healing, allogeneic tissue, decellularization protocol, regenerative medicine, fibroblasts, viability index

## Abstract

Background: Tissue regeneration is a complex process that allows wounds to heal. Many options are currently available to help human skin repair and to reduce the recurrence of hernias. The aim of this study is to analyze the best decellularization protocol for allogenic human dermal tissues. Methods: Dermal flaps from donors were used and compared with a control group. Each flap was subjected to seven different decellularization protocols and washed with a sequence of five solutions. The samples were then subjected to four control tests (such as Nile Red), and long-term contacts were analyzed to assess whether the decellularized dermis samples could support the growth of human fibroblasts. Results: All the samples had an average residual viability of 60%. Except for one sample, the decellularization treatments were able to reduce cell viability significantly. The Nile Red test showed a significant reduction in phospholipid content (mean 90%, *p*-value < 0.05) in all treatments. The cell growth increased in a linear manner. As described in the literature, sodium-dodecyl-sulfate (SDS) caused an interference between the test and the detergent. Conclusions: This paper shows the first step to finding the best decellularization protocol for allografting human dermal tissues. Further biocompatibility tests and DNA quantification are necessary.

## 1. Introduction

Permanent synthetic meshes have been used in clean wounds with excellent long-term results and low recurrence rates. Synthetic meshes are nonabsorbable meshes made of polypropylene (PP), expanded polytetrafluoroethylene, polyester, lightweight PP, or a combination of these materials to obtain ‘‘tension-free’’ closure in ventral hernia repairs; however, surgeons are reluctant to implant a permanent foreign material in a patient undergoing contaminated ventral hernia repair, principally because of the increased risk of postoperative infection, mesh extrusion, and fistula formation [1,2,3,4,5]. Current options have included staged repairs, primary fascial closure, or reinforcement with biologic mesh. The proposed advantage of a biologic mesh is that the patient’s immune cells can infiltrate the material to defend against the bacterial load and eventually replace the biologic mesh with the host tissue [6]. Unfortunately, the long-term durability of biologic grafts used for complex abdominal wall reconstruction has been disappointing [7,8]. Contaminated or infected hernia repair sites, reinforced with biologic scaffolds, are associated with recurrence rates as high as 50% [9]. Moreover, the cost of human and porcine-derived biologic meshes cannot be underestimated. One of the biggest prospective studies evaluating the performance of biosynthetic mesh in contaminated areas (the COBRA study) demonstrated that the biosynthetic mesh reinforcing midline fascial closures during the single-staged repair of contaminated ventral hernias had low hernia recurrence and postoperative wound infection rates. Moreover, the recurrence rate at 24 months remains high; 17% of patients suffer from a recurrence of a ventral hernia during follow-up appointments and 28% of patients present postoperative wound events such as surgical site infections, fistulas, wound dehiscence, hematomas, or seroma and bowel obstructions [10]. Using biosynthetic materials reduces the recurrence rate, but the biocompatibility of biologic meshes remains higher and safer. Moreover, a recent systematic review on hernias by Morris et al. turned the spotlight on the current clinical use of absorbable vs. Non-Absorbable Synthetic Meshes (NASM) in a contaminated area. The meta-analysis demonstrated the superiority of NASM over biologic and biosynthetic meshes regarding hernia recurrences (HR), surgical site infections (SSI), and unplanned reoperations. In such clinical scenarios, these countertrend results question the current consensus against NASM use, and encourage additional studies. Another essential factor to consider is the host tissue response to biologic materials, which plays a critical role in the constructive tissue remodeling of biologic scaffolds. Mononuclear cells, including proinflammatory (M1) and immunomodulatory and remodeling (M2) macrophages, are pivotal in terms of the host’s response to biological scaffolds. A high M2:M1 ratio favors constructive tissue remodeling over a chronic inflammatory response [11]. Host fibroblasts that successfully infiltrate the scaffold will proliferate and secrete components of the extracellular matrix, such as collagen and elastin. Low soft tissue distribution ratios of collagen I/collagen III have been shown to be associated with the failure of initial and repeat smooth tissue repairs, as collagen III confers less mechanical strength to tissue than collagen I. No studies have evaluated the potential of merging biosynthetic, biologic, or synthetic materials with the human dermis (hD). Moreover, in recent years, freeze-dried secretome from Mesenchymal Stem/Stromal Cells (MSCS) has been studied to promote cellular homing in vitro and in vivo. In previous years, the efficacy of MSCS derived from various connective tissues, for the purpose of enhancing bone regeneration through in vitro, in vivo, and clinical trials, has been extensively proven by many researchers [12]. A recent study, promoted by the University of Pavia, investigates the role of the MSC secretome, which is formulated as a ready-to-use and freeze-dried medicinal product (the Lyosecretome) to encourage the osteoinductive and osteoconductive properties of titanium cages. The in vitro evaluation showed a significant improvement in cell proliferation after 14 days. 

To achieve faster healing, an ideal wound dressing should be able to: provide a moist environment; enhance epidermal migration; promote angiogenesis and connective tissue synthesis; maintain correct blood flow to the wound bed; protect the wound bed against infection; stimulate debridement action to enhance leucocyte migration; be sterile, non-toxic, and non-allergic; and can be easily removed (or biodegradable) [13].

Traditional dressings fail to provide these characteristics, so they have been replaced by modern and advanced dressings. Nowadays, some materials deliver active components and interact directly with cells or specific chemicals in the wound site. In such scenarios, Negative Pressure Wound Therapy (NPWT) and Hyperbaric Oxygen Therapy (HBO) take place.

Today, with regard to advanced dressings, skin autografts are considered the gold standard for treating complex burns and chronic wounds. Skin autografts have most of the characteristics listed above. They do not have rejection issues, but some factors limit their use, such as location, the size of the wound’s area, patient’s general status, and most importantly, the availability of a free donor site; thus, skin substitutes represent an efficient alternative to skin grafts. Ideally, skin substitutes should replace skin, with permanent wound coverage, they should act as a barrier without toxicity or antigenicity, be resistant to hypoxia, be resistant enough to shear, and at the same time, be easy to apply and adjust to the irregularity of wound surfaces. In recent years, dermal scaffolds have found great application in covering wounds and supporting contemporary epidermal grafts. Especially in the last twenty years, research has focused on scaffolds derived from decellularized dermis, named acellular dermal matrices (ADMs). They have proven to be especially useful when treating chronic wounds because they allow a reduction in healing times. Driving the idea behind the introduction of decellularized dermis is the immunogenicity of allogeneic tissues and the fact that it comes from cellular antigenic components (nucleic acids, surface receptors) that trigger the recipient’s immune system. Moreover, the extracellular matrix (ECM) components are highly biocompatible, and they induce a moderate foreign body reaction that lasts for a short duration of time, meaning that it does not lead to rejection. The process aims to remove cells and components without altering the ECM composition. This has led to the development of several protocols and products which have high costs and different clinical results.

The REPAIR study aims to evaluate the in vitro cell migration and mesh cellularization in two different models: the first uses biosynthetic/synthetic mesh alone, and the second uses biosynthetic/synthetic mesh merged with human dermis that is freeze-dried with mesenchymal stem cell secretome (MSCS). The study design is summarized in Figure 1. This is the first paper that aims to determine the best decellularization protocol when obtaining human dermal tissue from tissue that has been donated. 

## 2. Materials and Methods

### 2.1. Dermal Flap Collection and Treatment

The dermal flaps came from 6 different donors (MI 44-21, MI 49-22, MI 51-21, MI 55-21, MI 58-21, MI 73-21), two females and four males, aged between 46 and 70 years (mean 56 ± 8.5, median 55). The site of sampling was the lumbar area and upper thighs. The thicknesses of the samples were categorized as thick (2 mm), middle (1.5 mm), and thin (1 mm). Differences between the samples’ thicknesses were influenced by the BMI and characteristics of the different donors. The samples were refined to obtain linear margins, and they were subsequently cut into flaps of 2 × 2 cm^2^. Each sample was then subjected to one decellularization protocol. Details about the composition of each protocol’s solutions are summarized and explained in Table 1. 

### 2.2. Decellularization Protocols

The decellularization protocols were selected according to the available data in the literature. The elimination of the cellular component results were more efficient with surfactants, meaning that CHAPS (3-[(3-Cholamidopropyl)dimethylammonio]-1-propane sulfonate hydrate, m.w. 614.9 g/mol, Sigma Aldrich) and SDS (Sodium dodecyl sulfate, 288.4 g/mol, Sigma Aldrich) seemed to be the most reliable and efficient. A pH = 8 was maintained in each solution to avoid the possibility of changing the architectural structure of the collagen and elastin fibers. To simplify and speed up the process, some protocols were built into the stock solutions in tubes. Details about stock solution preparation are reported in Appendix A. 

Tubes were then placed on a tilting plate (160 rpm), at 37 °C, for 16 h, except for the fifth protocol. The treatment for the fifth protocol involved using two different solutions, followed by the removal of the solution and overnight storage at 4 °C. Once the treatment was completed, the samples were washed with a sequence of five solutions (each test tube had 10 mL of solution added, and they were placed on a tilting plate at 160 rpm, at 37 °C). Details about each solution are reported in Appendix B. The 2 × 2 cm^2^ samples were then subjected to three decellularization control tests (Nile Red, MTT assay, and Hematoxylin/Eosin) and one modified extraction test, which investigate whether the decellularized flaps release residual degrading substances that are not compatible with cell growth.

### 2.3. Control Test

Nile Red (9-dietilammino-5H-benzo[α]fenossazin-5-one, Sigma Aldrich) is a vital dye that allows for quantifying the phospholipids, and consequently, it determines the residue of cell membranes, or part of them. A small sample was taken from one of the flaps, minced, and then digested with Type 1 A collagenase 2000 U (Sigma Aldrich) at 37 °C, for 24 h. The digested sample was filtered and then stained with Nile Red; the fluorescence was quantified with fluorimetry (excitation filter 520 nm; emission filter 610). MTT assay is a destructive colorimetric assay that allows the evaluation of cell viability. It is based on tetrazolium salts (44.5-Dimethyl-2-thiazolyl-2.5-diphenyl-2 H-tetrazolium bromide, Sigma Aldrich), which, after being incubated with the vital cells, are reduced to formazan due to the actions of the enzyme, succinate dehydrogenase, an enzyme present in mitochondria, and active only in vital cells. The reduced form was subsequently solubilized with DMSO (Dimethylsulphoxide, Sigma Aldrich), and the resulting colored solution was quantified by measuring absorbance at 670 nanometers using a spectrophotometer. For the Hematoxylin/Eosin control, each sample was placed in a vial containing 1 mL of 10% formalin (4% formaldehyde). The hematoxylin/eosin histological analysis was able to evaluate the presence or absence of cell residues following decellularization. 

### 2.4. Long-Term Culture Test

The aim was to assess whether the decellularized dermis samples could support human fibroblasts’ colonization and growth. Each of the decellularization protocols considered for the study obtained decellularized dermis flaps of 0.5 × 0.5 cm in size. The flaps were placed in the wells of an untreated 96-multiwell to limit the cells’ adhesion to the plastic support as much as possible. The same cutting procedure was followed for the gold standard (MTF 1.9 mm). Human fibroblasts (almost 5000 cells/well) were implanted on the decellularized dermis samples and MTF using DMEM HG soil. In the soil 10% FBS, 1% Glutamine, 0.5% Pen/Strep, and 0.5% Amphotericin B were added. Cell growth was evaluated at predetermined time intervals using the MTT test after 1, 7, 15, and 25 days of contact. Each sample was also analyzed histologically with the Hematoxylin and Eosin staining from day seven.

### 2.5. Statistical Analysis

Results were processed using StatGraphics XVII software (Statpoint Technologies, Inc., Warrenton, VA, USA). A generalized linear analysis of variance (ANOVA) model was used to study the efficiency of the decellularization process, considering the lipid content and metabolic activity as response variables, and the decellularization treatment and the donor as fixed factors. The most minor significance difference (LSD) test was used to estimate the differences between the means. A statistically significant result was set at *p* < 0.05. 

## 3. Results

To evaluate the effects of the decellularization protocols on the viability index (quantified by MTT assay), analysis was initially carried out on all the samples. It was expressed as optical density (OD) per mg (OD/mg). Three donors (MI44-21, 49-21, 51-21) were not subjected to decellularization due to the lack of starting tissue, and thus were excluded from the analysis. The samples were compared with a gold standard matrix (CTRL), taken from five decellularized matrices of different thicknesses (26 samples). The mean viability value of the controls was similar to the mean of the various treatments under examination (22.69 ± 1.55 OD/mg vs. 19.54 ± 2.28 OD/mg). Except for S0.5 and S1 treatments, all treated samples had an average residual viability of 60%, with a minimum of 53% for C2K, and a maximum of 66% for CS. Viability values and the number of samples are reported in Table 2. A one-way Anova test was then carried out in order to evaluate the effects of each protocol on viability. Results are reported in Figure 2, and they clearly show a similar mean residual viability for C1.5K, C1.5TE, C2TE, C2K, and CS protocols when compared with the CTRL group. The S1 and S0.5 protocols resulted in ‘over-the-mean’ residual viability, but as explained in the Section 4, this result depends upon interference between the MTT test and the SDS. 

A second test was carried out to confirm the data from the first analysis, only considering donors subjected to all decellularization protocols (MI 55-21; MI 58-21; MI 73-21) and excluding SDS data in order to reduce the possibility of a bias related to the interference between the SDS and MTT tests. The results of the two-way ANOVA analysis showed that both independent variables (donor and therapy) led to obtaining a *p*-value < 0.05 (Figure 2) in two of the three samples under evaluation (MI 58-21 and MI 55-21). During treatments which were equally applied to different donors, a specific variability between the OD/mg values can be seen in Table 3, Figure 2. This could be attributed to inter-individual factors and the variable thicknesses of the samples. It is confirmed that the decellularization treatments can significantly lower cell viability, except for the MI73-21 donor. The reason for this could be because the sample has a low cellular content due to the advanced age of the subject (70 years), and it has a considerably lower thickness than the other samples (1 mm MI 73-21; 2 mm MI 58-21; 2 mm MI 55-21). 

We then proceeded with the Nile Red test to evaluate decellularization’s effects on the phospholipid content. The first analysis was carried out considering all the samples (Table 4). When regarding the treatment as the only variable (Figure 3, Table 5), all the treatments tested showed a significant decrease (*p*-value < 0.05) in phospholipid content, with better results than the CTRL (CTRL 1.1 µg/mg ± 0.19; CS: 0.36 ± 0.30; C1.5TE 0.46 ± 0.26; C2TE 0.5 ± 0.26; C1.5K 0.40 ± 0.30; C2K 0.50 ± 0.30; S1 3.50 ± 0.30; S0.5 2.59 ± 0.26). In percentage terms, the treated samples have an average residual phospholipid content of 15%, with a minimum of 11% for CS, and a maximum of 17% for C2K. To confirm the data obtained, a second analysis was carried out to evaluate the effects of the treatment and the donor (Figure 4), which only took into account the donors that were subjected to all treatments, and it excluded SDS for the same reasons discussed above. These data confirmed the effectiveness of treatments in reducing the phospholipid content. 

Moreover, unlike the MTT assay, no differences were found between the non-treated samples in the Nile Red test. This could be explained by the fact that the models, in this case, were weight-normalized, thus reducing the thickness discrepancy between them. Data regarding the Nile Red test are summarized in Table 4 and Table 5 and are shown in Figure 4 and Figure 5. 

A third analysis was then performed, in which only the flaps that were subjected to all decellularization processes were considered (Figure 6). In this case, the statistical analysis showed significant differences between the donors’ values that were subjected to the same treatment. Notably, the phospholipid residue was significantly higher in the thicker sample (MI55-21). The main reason for this could be related to the difficulties in removing the residues in the thicker samples (although, they had the same cell content and were about the same weight). There were no substantial differences between medium and thin thickness samples. Overall, the analysis suggests that the most effective treatments to decrease the phospholipid content are CS, C1.5K, and C2K, especially in cases of high thickness. Finally, to further evaluate the reduction of the phospholipid content, the percentage of phospholipid residue was calculated. The comparison between the values of the medians shows a homogeneous distribution, in which the average value is equal to 10% (Figure 7); therefore, it can be stated that all treatments allow an average reduction of 90% with regard to the phospholipid content, with a maximum of 93% for CS. 

In addition to the quantitative analysis, a qualitative assessment of the tissue was also carried out in the form of pre- and post-decellularization treatments, and it was stained with H&E. The histological analysis was carried out entirely on only two donors (MI 49-21 and MI 55-21) and a control sample. Observing donor MI 49-21 (Figure 8), in the samples subjected to decellularization treatments, it is possible to note an effective decrease in the blue nucleic content. Compared with the untreated sample, these were devoid of viable cells, and they only show nuclear shadows (cell cytoplasm in the absence of a nucleus), together with persistent hair follicles. At the same time, the pink extracellular matrix remained intact, which is essential for obtaining a functional scaffold. The comparison with the control shows that SDS0.5, SDS1, and CS treatments were the most efficient in cell elimination. Donor MI 55-21 showed differences between the decellularization treatments (Figure 9), which was likely due to the greater thickness of the flap used. Observing the figures, it can be seen that the protocols with CHAPS still exhibit a relatively high nucleic content. In particular, solutions containing Tris-HCl/EDTATrisHCl/EDTA seem to be less efficient than those with KCl/HEPES, whereas the surfactant concentration does not affect the final result. With regard to the SDS-containing protocols, as with the previous donor, they provided noticeable results, and were comparable with the control sample. 

## 4. Discussion

Our study focuses on developing a decellularization protocol for the human dermis, which must ensure effective cell removal and reasonable maintenance of the integrity of the extracellular matrix. The decellularization of the dermis was carried out using chemical methods. All tested protocols achieved a good degree of decellularization. The analysis of the effects of the treatments on vitality using a one-way ANOVA test shows that the treatments applied significantly affected vitality (Table 2, Figure 2). For the S0.5 and S1 treatments, the observed values, compared with the untreated samples, do not seem to depend on the low effectiveness of the method, but on their level of interference with the MTT test. The work of N.T.H. Nga et al. [12] has highlighted how using a solution of SDS and DMSO leads to the MTT test becoming more sensitive, which results in obtaining a higher absorbance peak than the assay that was used solely with DMSO. This could also explain the average of 21.88 ± 2.28 OD/mg that was obtained using the CS protocol, which is slightly higher than the other treatments; however, a possible interference cannot be ruled out, even if the CS protocol has a lower concentration and the detergent was used at a different time than the previous ones. With regard to treatments with SDS, the hypothesis of interference between the test and the detergent reappears: the works of Keith Gooding et al. 1994 [13] and Indah Nurita Kurniasih et al. 2015 [14] highlighted the hydrophobic interactions that occur between the SDS in critical micellar concentration (CMC) and the Nile Red dye. The CMC of the surfactant is equal to 8.0 ± 0.03 mM [13]; the concentration is considerably lower than those used in the two protocols, at 17 mM (S0.5) and 34 mM (S1), respectively; therefore, the data relating to these protocols are not reliable. The CS protocol does not seem to be affected by this type of interaction as the concentration of SDS used in the treatment (3.5 mM) is lower than the CMC.

On a histological level, the best results were obtained with sodium dodecyl sulfate at concentrations of 0.5 and 1%; the resultant matrix was comparable to the commercial reference FlexHD^®^. Nevertheless, these protocols are unsuitable for human use because they were found to be toxic after 24 h of contact with cell cultures. On the other hand, the 0.1% sodium dodecyl sulfate solution, used concomitantly with 1% 3-[(3colaminopropyl) dimethylammonium]-1-propane sulfonate, allows the same histological results to be obtained, but it is free from toxicity in cell cultures. This difference is due to the permanence of the detergent inside the flap in the first two protocols because sodium dodecyl sulfate in critical micellar concentration can bind to collagen fibers. Regarding the use of 3-[(3colaminopropyl) dimethylammonium] -1-propane sulfonate, the protocols have shown similar behavior in the various analyses, without an evident increase in concentration; however, there seem to be differences, even if they are not statistically significant, based on the type of tampon used, and thus it will need to be investigated more thoroughly.

Furthermore, the samples subjected to treatment with this detergent did not show any interference with cell growth. Moreover, although the Nile Red test in our study appears to be the most promising, there are no data in the literature that establish the minimum phospholipid content necessary to avoid the immune reaction, nor to define whether tissue is decellularized. The only data available in the literature refer to the residual DNA content. In particular, it has been established [15] that the minimum criteria sufficient to satisfy the decellularization process are: less than 50 ng of DNA per mg of ECM dry weight; length of the DNA fragment less than 200 bp; and lack of visible nuclear material in tissue sections stained with DAPI or H&E.

This study has several limitations. First, the impossibility of performing a DNA-residual quantification test does not allow our findings to be completely reliable. On the other hand, our findings will be considered in our protocol’s second phase, and they could be a strong base for further in vitro studies in this field. Another critical issue is the variability of thickness in our dermal samples. It is well-known in the literature that cellular count is influenced by thickness.

For this reason, we performed weight normalization on the samples included in the analysis in the Nile Red test. Moreover, having three different thickness ranges has allowed us to understand and clarify the influence of donor-related factors in our protocols. This issue will be taken into account in all future phases of this project. 

## 5. Conclusions

In conclusion, future developments should be focused on identifying a more effective washing method for removing sodium dodecyl sulfate from the scaffolds. Moreover, biocompatibility tests will be performed via contact and direct seeding on the matrix. A DNA quantification will also be carried out because the only data available in the literature presents minimum criteria that are sufficient to satisfy the decellularization process.

## Figures and Tables

**Figure 1 jpm-12-01411-f001:**
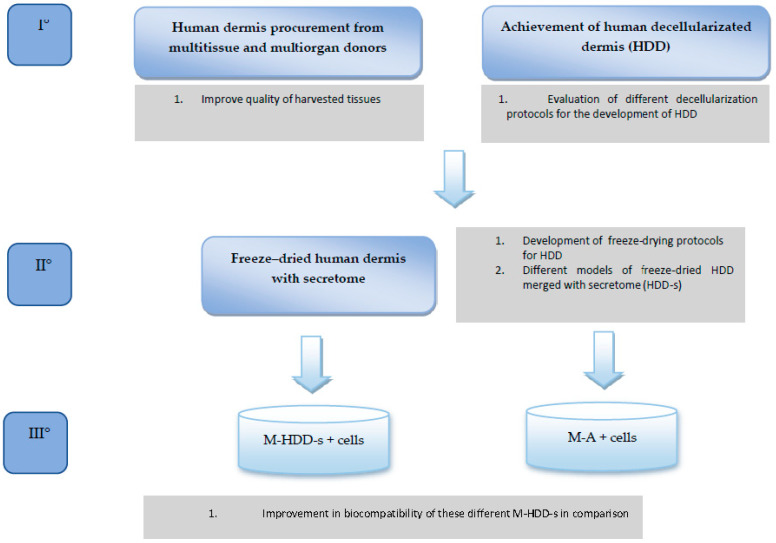
REPAIR study protocol.

**Figure 2 jpm-12-01411-f002:**
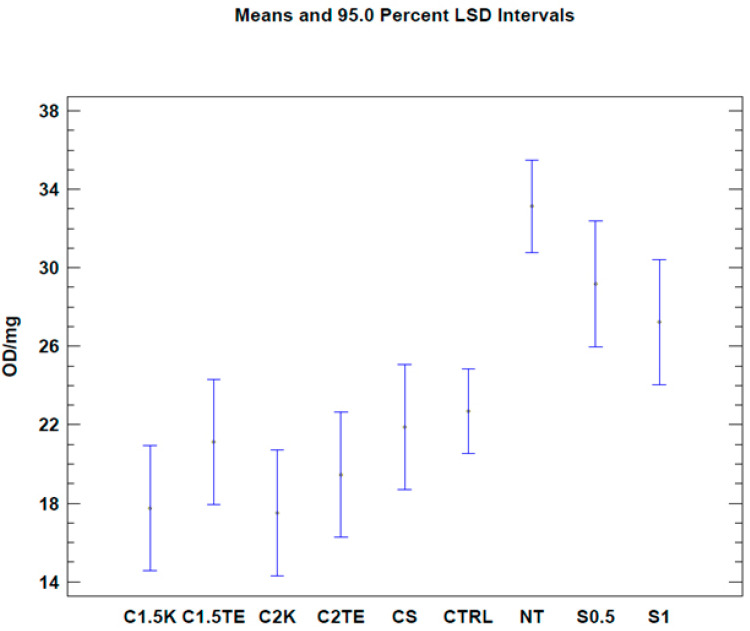
One-way ANOVA test. Mean of viability expressed as OD/mg for each protocol. NT: not treated. CTRL: control samples.

**Figure 3 jpm-12-01411-f003:**
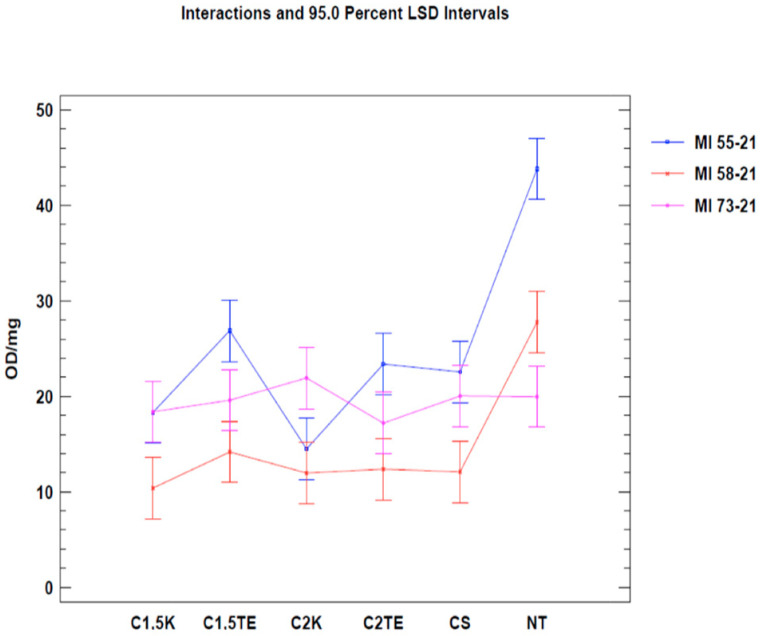
Two-way ANOVA analysis: mean of OD/mg according to donor and treatment.

**Figure 4 jpm-12-01411-f004:**
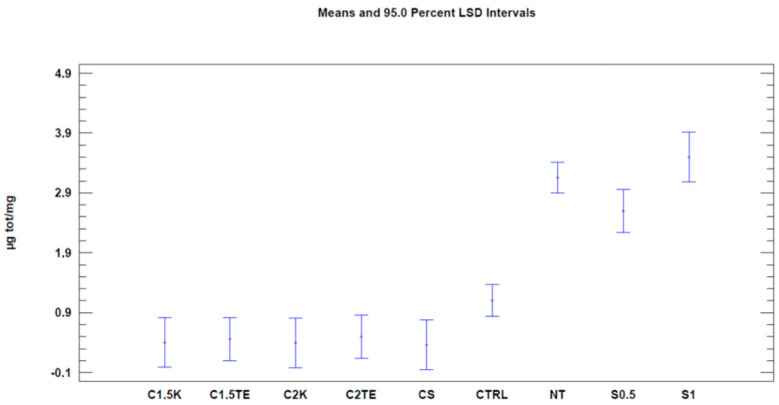
One-way ANOVA analysis: mean of µg/mg for each treatment.

**Figure 5 jpm-12-01411-f005:**
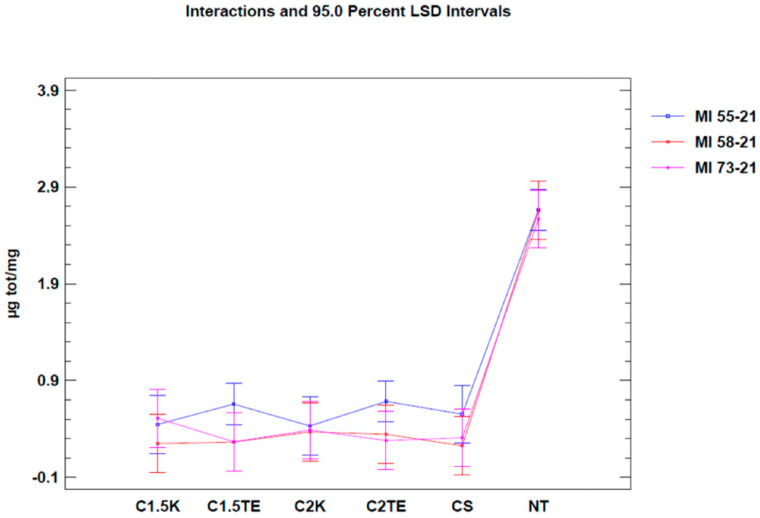
Two-way ANOVA analysis: mean of µg/mg according to donor and treatments.

**Figure 6 jpm-12-01411-f006:**
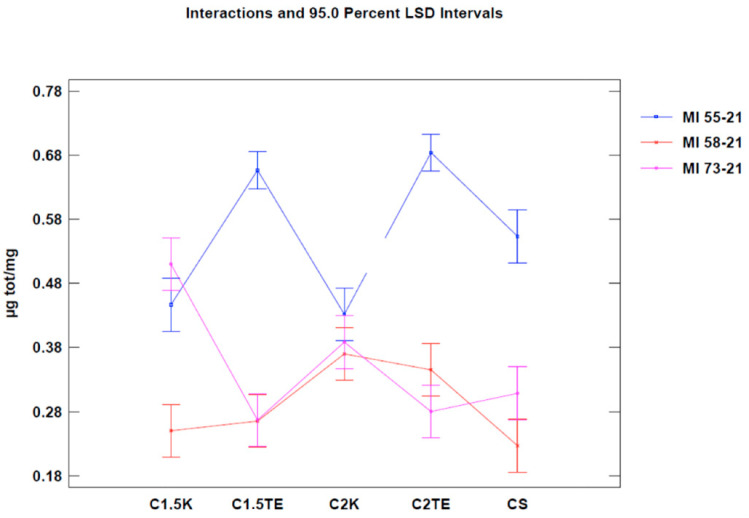
Two-way ANOVA analysis: mean of µg/mg of donor-decellularization treatments.

**Figure 7 jpm-12-01411-f007:**
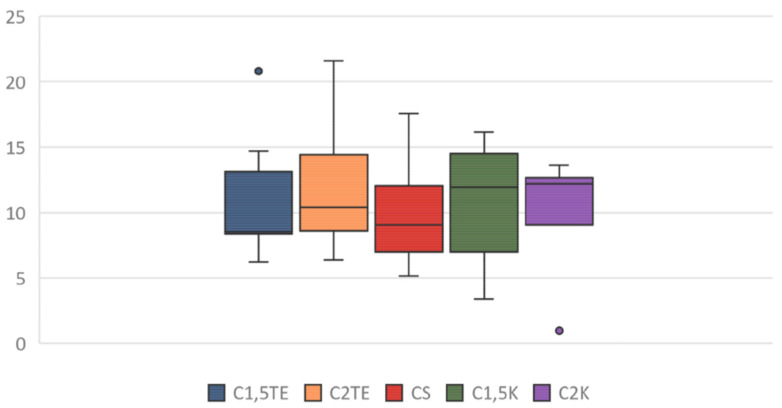
Box plot: percentage of phospholipid residue after treatment application.

**Figure 8 jpm-12-01411-f008:**
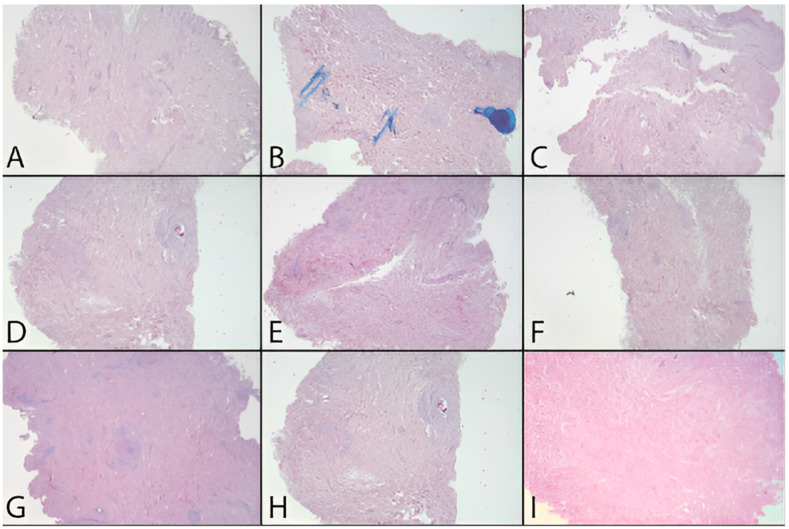
H&E dermis MI 49-21 treated and before treatment. (**A**) Before treatment, (**B**) C1.5TE, (**C**) C2TE, (**D**) C1.5K, (**E**) C2K, (**F**) CS, (**G**) S0.5, (**H**) S1, (**I**) MTF.

**Figure 9 jpm-12-01411-f009:**
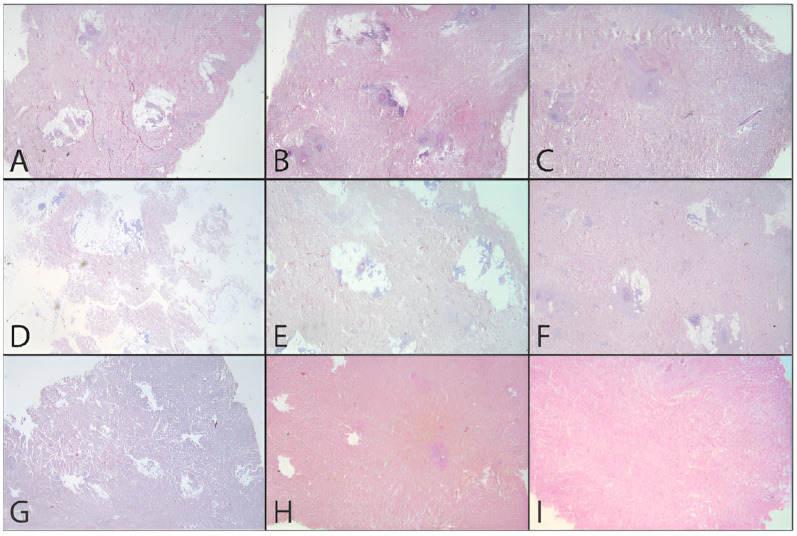
H&E dermis MI51-21 treated and before treatment. (**A**) Before treatment, (**B**) C1.5TE, (**C**) C2TE, (**D**) C1.5K, (**E**) C2K, (**F**) CS, (**G**) S0.5, (**H**) S1, (**I**) MTF.

**Table 1 jpm-12-01411-t001:** The seven decellularization protocols used in the study and the components’ concentrations. CHAPS (3-[(3-Cholamidopropyl)dimethylammonio]-1-propanesulfonate hydrate, Sigma Aldrich) (Milan, Italy); SDS 10% (Sodium dodecyl sulfate, Sigma Aldrich); NaCl 5 M; Buffer Tris-HCl/EDTA (TE) 1 M/0, 1 M at Ph 8; HEPES (4-(2-hydroxyethyl)-1-piperazineethanesulfonic acid, Sigma Aldrich).

Protocol Name	Composition
C1.5TE	CHAPS 1.5%
	Tris-HCl 75 mM (157.6 g/mol)
	NaCl 165 mM (58.44 g/mol)
	EDTA 7.5 mM (292.24 g/mol)
C2TE	CHAPS 2%
	Tris-HCl 100 mM
	NaCl 220 mM
	EDTA 10 mM
S0.5	SDS 0.5%
	Tris-HCl 25 mM
	NaCl 75 mM
	EDTA 2.5 mM
S1	SDS 1%
	Tris-HCl 50 mM
	NaCl 150 mM
	EDTA 5 mM
CS	**First solution:**
	CHAPS 1%
	Tris-HCl 50 mM
	NaCl 110 mM
	EDTA 5 mM
	**Second solution:**
	SDS 0.1%
	Tris-HCl 5 mM
	NaCl 15 mM
	EDTA 0.5 mM
C1.5K	CHAPS 1.5%
	KCl 2.25 M (74.55 g/mol)
	HEPES 0.75 M (238.30 g/mol)
C2K	CHAPS 2%
	KCl 3M
	HEPES 1M

**Table 2 jpm-12-01411-t002:** Mean of OD/mg for each treatment performed on all samples under evaluation and CTRL.

*Treatment*	N° of Samples	Mean OD/mg	Standard Deviation
NT	22	33.14	1.68
C1.5TE	12	21.12	2.28
C2TE	12	19.45	2.28
S0.5	12	29.17	2.28
S1	12	27.23	2.28
CS	12	21.88	2.28
C1.5K	12	17.74	2.28
C2K	12	17.5	2.28
CTRL	26	22.69	1.55
**Total**	132		

**Table 3 jpm-12-01411-t003:** Mean of OD/mg fo.

*Donor* Treatment	N° of Sample	Mean OD/mg	Standard Deviation
MI 55-21 NT	3	43.82	2.24
MI 55-21 C1.5 TE	3	26.88	2.24
MI 55-21 C2 TE	3	23.38	2.24
MI 55-21 CS	3	22.57	2.24
MI 55-21 C1.5K	3	18.31	2.24
MI 55-21 C2K	3	14.5	2.24
MI 58-21 NT	3	27.76	2.24
MI 58-21 C1.5 TE	3	14.17	2.24
MI 58-21 C2 TE	3	12.37	2.24
MI 58-21 CS	3	12.09	2.24
MI 58-21 C1.5K	3	10.4	2.24
MI 58-21 C2K	3	11.96	2.24
MI 73-21 NT	3	19.96	2.24
MI 73-21 C1.5 TE	3	19.59	2.24
MI 73-21 C2 TE	3	17.2	2.24
MI 73-21 CS	3	20.04	2.24
MI 73-21 C1.5K	3	18.39	2.24
MI 73-21 C2K	3	21.92	2.24
**Total**	54		

**Table 4 jpm-12-01411-t004:** Mean of µg/mg for each treatment.

Treatment	N° of Sample	Mean of µg/mg	Standard Deviation
NT	48	3.16	0.18
C1.5TE	24	0.46	0.26
C2TE	24	0.5	0.26
S0.5	24	2.59	0.26
S1	18	3.5	0.3
CS	18	0.36	0.3
C1,5K	18	0.4	0.3
C2K	18	0.5	0.3
CTRL	45	1.1	0.19
**Total**	237		

**Table 5 jpm-12-01411-t005:** Mean of µg/mg for donor treatment.

*Donor* Treatment	N° of Samples	Mean µg/mg	Standard Deviation
MI 55-21 NT	12	2.66	0.15
MI 55-21 C1.5 TE	12	0.66	0.15
MI 55-21 C2 TE	12	0.68	0.15
MI 55-21 CS	6	0.55	0.21
MI 55-21 C1.5K	6	0.45	0.21
MI 55-21 C2K	6	0.43	0.21
MI 58-21 NT	6	2.66	0.21
MI 58-21 C1.5 TE	6	0.27	0.21
MI 58-21 C2 TE	6	0.35	0.21
MI 58-21 CS	6	0.23	0.21
MI 58-21 C1.5K	6	0.25	0.21
MI 58-21 C2K	6	0.37	0.21
MI 73-21 NT	6	2.57	0.21
MI 73-21 C1.5 TE	6	0.27	0.21
MI 73-21 C2 TE	6	0.28	0.21
MI 73-21 CS	6	0.31	0.21
MI 73-21 C1.5K	6	0.51	0.21
MI 73-21 C2K	6	0.39	0.21

## Data Availability

For all presented data please refer to michele.altomare@uniroma1.it.

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
