# Peer review of "Regenerative Medicine to Improve Outcomes of Ventral Hernia Reconstruction (REPAIR Study) Phase 1: Find the Best Decellularization Protocol for the Human Dermis"

_jpm, 2022, doi:10.3390/jpm12091411_

Round 1

Reviewer 1 Report

Altomare et al. present an important point on using decellularized scaffolds over synthetic for reconstructive surgeries. The author aims to identify the best decellularization protocol for the human dermis. However, the manuscript needs a major revision. The authors need a clearer characterization of the decellularized scaffolds that include higher resolution H&E images and a clear description of the methods and images presented. The result section with a clear data presentation can also help the readers better understand the author’s work. 

Major critics:

The method section is not clearly written, and the authors should define solutions such as the surfactant solution used on page 4. On pages 5 and 6, the description of the MTT assay is repetitive. 

The authors need to clearly characterize the decellularized samples at the beginning of the results. The H&E images are not clear and of poor quality. Images with higher resolution and included higher magnifications are needed. The authors state on page 6 that they evaluated the effects of the decellularization protocols on the viability index (quantified by MTT assay), which is important; however, a clearer characterization of the decellularized sample is initially needed. The authors need to include a clearer H&E before (if possible) and after decellularization for multiple samples. Then followed by the Nile Red test, then an analysis of the DNA content left behind because, as the authors suggest, this is one criterion used to assess proper decellularization.   

The paper has multiple tables and graphs that are unclear. The images need clear figure legends and clear X and Y axis labels. Higher resolution images are also necessary to understand the presented data, and some tables can also be combined for clearer data presentation. Description of the composition abbreviation placed under Table 1 can help readers understand the components.

Grammatical error correction is also needed.

Author Response

Dear Reviewers, 

Thank you very much for your suggestions. 

Please find attached the point-to-point letter of response. 

Yours sincerely, 

Michele Altomare

Reviewer 2 Report

1. Reference 12 starts with ‘4.’, a typing error, which should be erased. Besides, even though the Lyosecretome study the authors mention in the next sentence may be an easy one to find in the literature, the authors should cite the work properly.

2. In the tables and graphs, commas were preferred by the authors. C1.5K is present on the text whereas C1,5K is present on graph axis labels. As such, the authors need to decide on either point or comma for consistency. Decimal operator should be chosen as point instead of comma.

3. Some of the figures have horizontal lines as if they are captured from a low-resolution screen. Can the authors resolve this issue? Also, in some figures containing microscope images such as Figure 10, white balances of the images are not consistent, creating distraction to the reader. Can the authors capture and save new images with consistent white balances if possible? Before capturing each image, setting the white balance automatically from the software of the microscope may be done to take consistent microscope images.

4. Why does Table 2 come after Table 6 in the manuscript? Table 7 and Table 8 are not present in the manuscript but they both are referred in the text. 

Author Response

(The authors gave the same response as above.)

Round 2

Reviewer 1 Report

The authors have made sufficient changes to the manuscript.